# Dietary Natural Compounds and Vitamins as Potential Cofactors in Uterine Fibroids Growth and Development

**DOI:** 10.3390/nu14040734

**Published:** 2022-02-09

**Authors:** Iwona Szydłowska, Jolanta Nawrocka-Rutkowska, Agnieszka Brodowska, Aleksandra Marciniak, Andrzej Starczewski, Małgorzata Szczuko

**Affiliations:** 1Department of Gynecology, Endocrinology and Gynecological Oncology, Pomeranian Medical University, 71-252 Szczecin, Poland; jolanta.nawrocka.rutkowska@pum.edu.pl (J.N.-R.); agabrod@wp.pl (A.B.); o.marciniak@wp.pl (A.M.); andrzejstarcz@o2.pl (A.S.); 2Department of Human Nutrition and Metabolomics, Pomeranian Medical University, 71-460 Szczecin, Poland; malgorzata.szczuko@pum.edu.pl

**Keywords:** uterine fibroids, diet, green tea, curcumin, vitamin A, C, D, E, selenium, trace elements

## Abstract

An analysis of the literature generated within the past 20 year-span concerning risks of uterine fibroids (UFs) occurrence and dietary factors was carried out. A link between Vitamin D deficiency and UFs formation is strongly indicated, making it a potent compound in leiomyoma therapy. Analogs of the 25-hydroxyvitamin D3, not susceptible to degradation by tissue 24-hydroxylase, appear to be especially promising and tend to show better therapeutic results. Although research on the role of Vitamin A in the formation of fibroids is contradictory, Vitamin A-enriched diet, as well as synthetic retinoid analogues, may be preventative or limit the growth of fibroids. Unambiguous conclusions cannot be drawn regarding Vitamin E and C supplementation, except for alpha-tocopherol. Alpha-tocopherol as a phytoestrogen taking part in the modulation of estrogen receptors (ERs) involved in UF etiology, should be particularly avoided in therapy. A diet enriched in fruits and vegetables, as sources of carotenoids, polyphenols, quercetin, and indole-3-carbinol, constitutes an easily modifiable lifestyle element with beneficial results in patients with UFs. Other natural substances, such as curcumin, can reduce the oxidative stress and protect against inflammation in leiomyoma. Although the exact effect of probiotics on uterine fibroids has not yet been thoroughly evaluated at this point, the protective role of dairy products, i.e., yogurt consumption, has been indicated. Trace elements such as selenium can also contribute to antioxidative and anti-inflammatory properties of a recommended diet. In contrast, heavy metals, endocrine disrupting chemicals, cigarette smoking, and a diet low in antioxidants and fiber were, alongside genetic predispositions, associated with UFs formation.

## 1. Introduction

Uterine leiomyomas are the most frequent tumors in women at reproductive age. The origin of these benign neoplasms is multifactorial. Genetic, inflammatory, hormonal, and other associated factors play an important role in uterine fibroids (UFs) development. Molecular analysis of this type of tumors points towards the mediator complex subunit 12 (*MED12*) mutations and high mobility group AT-hook 2 (HMGA2). The above genetic variants have different gene expression profiles [1,2,3]. The diversity of metabolic processes and reduced levels of specific vitamins and other co-factor metabolites enable tumor growth and formation. This process is conditioned by alterations in enzyme function and signaling pathways [4]. *MED12* mutations are associated with the induction of gene expression of wingless-type mouse mammary tumor virus integration site family, member 4 (Wnt4), and activation of β-catenin signaling in UFs [5]. The activation of mammalian target of rapamycin (mTOR) pathway can also be responsible for the growth of fibroids [6]. Oxidative stress has also been indicated as a potential factor [7]. Profibrotic tumors can form a result of the imbalance between connective tissue production and degradation. Oxidative stress is involved in the deposition of extracellular matrix (ECM) components in myoma, such as collagen, proteoglycan, and fibronectin [8]. Local and general inflammatory status may also present as a possible mechanism underlying the onset of myomas [9,10,11]. There is a growing body of knowledge on the management and conservative treatment of UFs. Dietary components can alter various factors associated with uterine myoma formation and have both promoting and inhibiting effects on tumor formation and growth [12,13,14,15,16,17,18,19,20,21].

The most popular, that is hormonal and surgical treatment of myoma, can have many potential side effects and complications. Thus, the trend of finding less invasive and more natural methods of managing leiomyoma appears to be very important in the era of postponement of patients’ procreation plans and the growing problem of infertility. In the era of increased interest in organics, environmentalism, and ever changing patient expectations, natural substances represent a promising strategy of action in the prevention, treatment, and management of fibroids and, as such, may provide complementary therapy for this common pathology.

### 1.1. Nutrients and Their Deficiencies

A recent study by Makwe et al. revealed lower serum levels of vitamin C, vitamin D, and calcium in black women with uterine fibroids [22]. The hypothesis was that these vitamins and minerals played a role in etiopathogenesis, progression, and growth of UFs. In a study by Orta et al., the effect of frequent dairy consumption was highlighted as a factor that might influence the incidence of uterine leiomyoma [13]. Despite no straightforward correlation between dairy consumption and fibroid formation, the authors noticed that a greater intake of yogurt and calcium-rich dairy could inversely reduce the risk of fibroids occurrence. Similar observations were made by Shen et al. and Zhou et al. who claim that dairy consumption may indeed have preventative effects when it comes to UFs formation [15,17]. Wise et al. observed an inverse association between dairy consumption and uterine leiomyomata development in African American women [20,21]. Since the dairy consumption is significantly lower among Black Americans than White Americans, they attributed dietary habits as a possible factor contributing to the frequency of the disease prevalence by race [21]. This may be related to the microflora of dairy products and sources of calcium in food alike. Dairy can be considered not only as a food source supporting the development of normal bacterial flora (probiotics, prebiotics, intestinal passage support), but also as the supplier of compounds and antioxidants.

Fruit and Vegetables are rich in dietary fibers, phytochemicals, vitamins and minerals, and antioxidants. These phytochemicals have well-known anti-inflammatory, antiproliferation, antifibrotic, and anti-vascular properties [23], thus in light of current knowledge, a diet based on a large amount of plant and dairy products seems to be a favorable recommendation. The focus of our study was on natural compounds that protect against the risk of UFs and can be useful in treatment of these tumors. After an extensive review of the available Literature on the effects of dietary components on the occurrence of UFs (past two decades), we hypothesized that foods with proven antioxidant, anti-inflammatory, antiproliferative, and antifibrotic properties might be considered a potential treatment for patients with uterine myomas. All the studies discussed in the review are presented in Appendix A.

### 1.2. The Influence of Intestinal Dysbiosis

Dysbiosis, or dysregulation of the gut microbiota, may play a role in the pathogenesis and perpetuation of inflammatory processes [24]. Dairy products, as a source of vitamins and minerals, may potentially reduce the inflammation and tumor growth and thus have a beneficial effect on leiomyoma occurrence and growth. Some of these products, produced by bacterial fermentation of milk, i.e., yogurts, may change intestinal microbiota in consumers. Short chain fatty acids (SCFA) are produced in the gut as bacterial metabolites [25]. Butyric acid (BA), one of SCFAs, can induce differentiation and apoptosis and inhibit proliferation and angiogenesis [26]. Dysbiosis may stimulate the pro-inflammatory cytokines or growth factors. Growth factors and cytokines interact through the estrogen and progesterone action, which plays an important role in uterine leiomyoma growth [27]. A beneficial gut microbial environment can potentially affect the uterus environment and reduce the risk of uterine leiomyoma formation [28]. The gut microbiota covers the entire population of microorganisms: bacteria, fungi, viruses, and protozoa that exist in symbiosis with the human gut. Probiotics are living microorganisms that have a positive effect on gut microbiota. Changes in gut microbiota associated with probiotics intake have been shown to have positive effects in a number of diseases, mainly associated with chronic inflammation and oxidative stress [29,30,31,32,33,34], but to date, there have not been many studies on probiotic impact on uterine leiomyoma—only an indirect role of intestinal microflora changes resulting from yoghurt consumption and their effect on myoma formation has been observed [13,17,20,21].

## 2. Vitamins

Particular attention should be paid to carotenoids and Vitamin A derivatives. Carotenoids sourced directly from diet actively decrease reactive oxygen species and thus the oxidative stress response in tissues. Carotenoids are a major source of vitamin A in a diet. Lycopene, for instance, displays antioxidant properties and provitamin A activity, and is contained in many yellow, orange, and red fruits and vegetables [12,14,15,35,36]. In other words, carotenoids may play a role in diminishing the number and size of leiomyomas by its antioxidant effect. They cause suppression of cell proliferation and induce cells differentiation and apoptosis [16]. Vitamins C and E, as antioxidants, protect cell membranes and the DNA from oxidative stress, and vitamin A is essential for cell differentiation and proliferation control and may help reduce fibroid growth [18].

Retinoids, as derivatives of Vitamin A, have structural or functional similarity to vitamin A. Retinoids can be natural or synthetic. The proven effects of retinoids include reduction in inflammation, regulation of cell growth and proliferation, and inhibition of carcinogenesis. Studies confirm that retinoids inhibit the growth of primary cultures of human uterine myomas [37,38,39,40,41]. Tomatoes and tomato-based products are particularly high in lycopene, folate, vitamin C, vitamin A, and flavonoids. These bioactive compounds, as potent antioxidants, can be used with therapeutic effects in leiomyoma treatment. Changes in diet may have more beneficial effects than selective supplementation.

### 2.1. Carotenoids, such as Lycopene

Carotenoids as antioxidants can potentially reduce the risk of uterine leiomyoma. He et al. argued that a consumption of fruit and vegetable products rich in fibers and lycopene, significantly decreased the risk of fibroids in premenopausal women, while in postmenopausal women that correlation was insignificant [14]. This was confirmed in Shen Y. et al. 2016 study [15]. They revealed that a vegetarian diet, with greater intake of fresh fruits (i.e., apples) and vegetables (i.e., cruciferous vegetables, especially cabbage, Chinese cabbage, broccoli, and tomatoes rich in lycopene) was likely to significantly reduce the incidence of UFs. Similarly, Sahin et al. observed in their animal model study that high doses of lycopene supplementation in a form of tomato powder could prevent development and/or cause shrinkage of fibroids [35,36]. The doses of lycopene used by Sahin et al. were however much higher than in a typical human diet. Zhou et al. noticed that the risk of UFs formation could be significantly decreased with increased nut and vegetable consumption (especially legumes, seaweed, and carrots). Their study found no significant correlation with fruit intake (except for kiwi), which was inversely associated with the risk of UFs [17]. In contrast, Wise et al. showed that citrus fruit consumption was inversely associated with fibroid risk among Black American women [16]. Martin et al. suggested a positive but not statistically significant association between b-carotene and UFs [18]. Some studies do not support the hypothesis of an exclusively beneficial role of carotenoids in the risk of myoma. No association was found between UF risk and dietary intake of carotenoids (e.g., α- and β-carotene, lycopene: tomato juice, spaghetti, watermelon, salad greens, carrots, spinach, sweet potatoes, greens) in the 2021 study by Wise et al., confirming previous results published in 2011 [12,16]. Terry et al. observed a similar lack of correlation, stressing that cigarette smoking combined with high β-carotene intake exacerbated the risk of fibroid formation [19]. Czeczuga-Semeniuk analyzed the presence of different types of carotenoids (β-carotene, β-cryptoxanthin, lutein, neoxanthin, violaxanthin, and mutatoxanthin) in tissue of a healthy uterus and various uterine tumors, including myomas [42]. Lutein epoxide and mutatoxanthin were predominant in myomas. This points toward the fact that carotenoids with provitamin A activity may induce fibroid growth.

Dietary intake of carotenoids raises many questions with regard to their effects on fibrotic tumors, especially in female patients who smoke cigarettes, and as such requires further study.

### 2.2. Vitamin A. Retinoids

A 2020 study by Wise et al. found no association of UF incidence with dietary Vitamin A intake (i.e., salad greens, carrots, spinach, sweet potatoes, eggs, cheese, and cereal products) [12]. These results were generated within a particular racial group of women and as such might be worth exploring in the remaining population. Previously, in 2011, Wise et al. observed an inverse correlation between dietary intake of Vitamin A and the risk of UFs. They noted that it was predominantly conditioned by preformed Vitamin A derived from animal sources (i.e., liver and milk), but not by provitamin A from fruit and vegetable sources (i.e., carrots, sweet potatoes/yams, and collard greens) [16]. These findings may complement the molecular study of Heinonen et al. who showed that levels of vitamin A were specifically reduced in leiomyomas of the *MED12* subtype [4], yet appeared to be contradictory to conclusions drawn by Martin et al. who noted positive, statistically significant, and dose-dependent association between vitamin A and uterine fibroid risk, except for the population of Hispanic women [18].

A possible explanation for the positive association between Vitamin A and the risk of UFs is that exposure to high levels of Vitamin A can activate peroxisome proliferator-activated receptors (PPARs). These nuclear receptors, in combination with retinoid X receptors, can activate gene expression, and thus simultaneously increase the risk of UFs in some women [18]. Retinoids are small molecule derivatives of Vitamin A. Studies have shown that the retinoid pathway is significantly altered in fibroids compared to normal myometrium. Retinol requires conversion to retinoic acid (RA), which requires the activity of specific enzymes. In fibroid fibroblasts, aldehyde dehydrogenase (ALDH1) has been specifically identified as one of them. Studies concluded that alterations in the retinoid pathway could lead to abnormal RA production and signaling, which might be important in fibroid development [38,39,43,44]. Zaitseva reports that transcription factor II (known as NR2F2) and CTNNB1(b-catenin) genes are potentially causal factors in the development of UFs [45]. According to her research, the combination of RA and progesterone regulates NR2F2 expression and thus affects fibroid growth. This suggests that retinoids, by causing these molecular alternations, may be useful in the treatment of UFs. In their in vitro studies, Ben-Sasson et al. and Malik et al. demonstrated decreased cell proliferation, ECM formation, RA metabolism, transforming growth factor beta (TGF- β) regulation, and increased apoptosis in human leiomyoma treated with retinoic acid (ATRA-all-trans-retinoic acid) [40,41]. Broaddus et al. showed, again in vitro, that treatment with 4-(N-hydroxyphenyl)-retinamide (4-HPR) or a-difluoromethylornithine (DFMO) resulted in growth inhibition of primary cultures of human uterine leiomyoma [37]. 4-HPR is a synthetic retinoid analog that, in comparison to other retinoids, has a reduced toxicity potential. It promotes apoptosis and induces growth inhibition through induction of p53, p21, and p16, and modulation of extracellular matrix in human uterine leiomyomas. It was noted that the mechanisms of growth inhibition and apoptosis induction could be independent of binding to nuclear retinoid receptors [37].

The above results suggest that a diet rich in Vitamin A and retinoids can prevent fibroids and inhibit tumor growth. Synthetic retinoid analogues can also be effective.

### 2.3. Vitamin E

Vitamin E is a potent antioxidant that acts by scavenging lipid hydroperoxyl radicals and so can protect cells from the effects of free radicals [46]. Food sources of Vitamin E include canola oil, olive oil, almonds and peanuts, meat, dairy, leafy greens, and fortified cereals. It is also available in oral supplements. Little data is available on Vitamin E and its effects on UFs. Wise et al. found no associated risks with consumption of diet-derived Vitamin E [16]. Martin et al. highlighted a positive, dose-dependent link between vitamin E and the incidence of UFs; the findings were, however, not statistically significant [18]. Ciebiera et al. showed higher serum concentration levels of α-tocopherol (the most common form of vitamin E) in Caucasian women, which may be an important factor in fibroid development [47].

Vitamin E, despite its antioxidant properties, appears not to demonstrate proven beneficial effects in terms of leiomyoma prevention and management.

### 2.4. Vitamin D

Vitamin D is obtained mainly from sun exposure (skin synthesis), food (oily fish such as trout, salmon, tuna, mackerel, and fish liver oils), and vitamin supplements. Prohormonal forms of vitamin D require hydroxylation in the liver to 25-hydroxyvitamin D (25(OH)D) and in the kidney to their active form, that is 1,25-dihydroxyvitamin D (1,25(OH)_2_D_3_). Vitamin D_3_ exerts its biological functions by interacting with and activating the nuclear vitamin D receptor (VDR). In vitro studies point toward uterine myoma cells exhibiting lower levels of VDR expression [48,49]. In addition, a negative correlation between decreased levels of vitamin D receptor (VDR) and increased levels of estrogen and progesterone receptors (ER-α, PR-A, PR-B) was observed in myoma tumors [50]. It is estimated that Vitamin D deficiency affects 25–50% (possibly more) of patients [51]. Several studies found that insubstantial levels of Vitamin D can contribute to the development of UFs in African American, Caucasian, and Asian women alike [52,53,54,55,56,57,58,59,60]. Vitamin D_3_ deficiency activates fibroid cell growth, exacerbates DNA damage, and reduces DNA repairability; it promotes uncontrolled proliferation and fibrosis, and increases chronic inflammation. In combination, these processes are highly tumorogenic [61,62]. Othman et al. demonstrated that, in comparison to normal myometrium, myoma tissue contains significantly lower concentrations of 1,25(OH)_2_D_3_. Additionally, an overexpression of 24-hydroxylase was found in myoma, which may further suppress the anti-tumor effect of 1,25(OH)_2_D_3_ and exacerbate vitamin D deficiency in the tissue [63]. A recent study by Ciebiera et al. revealed an inverse correlation between lower 25(OH)D serum concentrations and increased serum transforming growth factor β3 (TGF-β3) concentrations in women affected by fibroids [64]. This growth factor can be associated with increased fibrosis and ECM accumulation in myoma [65,66]. Findings on inverse correlation between serum levels of 25(OH)D and fibroid volume vary, ranging from significant to insignificant association [52,67,68]. No correlation was however observed between 25(OH)D serum levels and number of fibroids [67]. Some data suggest that Vitamin D supplementation reduces leiomyoma cell proliferation and thus prevents leiomyoma growth [56,69,70,71]. A significant downregulation of ER-α, PR-A, PR-B, and steroid receptor coactivators in human myoma cells may be one of the mechanisms—an effect similar to that observed during the course of hormone therapy with GnRH analogues and ulipristal acetate (UPA) [50,72,73]. Halder et al. and Li et al. point to the antifibrotic activity of Vitamin D [74,75]. Vitamin D_3_ inhibited TGF-β3-induced protein expression and all TGF-β3-mediated effects involved in the fibrotic processes in leiomyoma. Other studies indicate that increasing Vitamin D levels by one unit can reduce the risk of developing UFs by 4–8% [59,64]. Hajhashemi et al. confirmed a significant decrease in leiomyoma size after 10 weeks of vitamin D administration [76]. A slight, statistically insignificant reduction in fibroid volume after a short-term Vitamin D supplementation was observed by Arjeh et al. [77], Davari Tanha et al. [78], and Suneja et al. [79] in patients with hypovitaminosis D (12, 16, and 8 weeks, respectively). Ciavattini et al. reported a similar effect after 12 months of Vitamin D_3_ supplementation [67]. Vitamin D_3_ supplementation may inhibit the growth of UFs, reduce fibroid-related symptoms, and reduce the need for surgical or medical treatment for progression of fibroids [67,78,79]. Especially, a long-term course of treatment can have antiproliferative, antifibrotic, and proapoptotic effects in leiomyoma, as demonstrated by Corachán et al. [80], a finding consistent with other studies in vitro [61,74,75]. Beside apoptosis induction, Vitamin D suppresses catechol-O-methyltransferase (COMT) expression and activity in myoma cells—an enzyme that plays a vital role in myoma formation [74]. In fact, physiological concentrations of vitamin D can effectively inhibit the growth of myoma cells [61]. 1,25(OH)_2_D_3_ can significantly reduce the expression of ECM-associated proteins and structural actin fibers in human leiomyoma cells, as observed by Halder et al. [48]. This effect was a consequence of previous significant induction of nuclear vitamin D receptor (VDR) expression by 1,25(OH)_2_D_3_ in a concentration-dependent manner. In another study, Halder et al. observed a significant reduction in MMP-2 and MMP-9 mRNA levels, as well as a reduction in MMP-2 and MMP-9 protein levels in uterine fibroid cells in a concentration-dependent manner and concluded that through this mechanism, 1,25(OH)_2_D_3_ might limit fibroid growth and ECM deposition [81]. Al-Hendy et al. observed that 1,25(OH)_2_D_3_ spontaneously induced its own VDR, while significantly downregulating the expression of sex steroid receptors (ERs and PRs) and receptor coactivators, which affected myoma formation and growth; hence, 1,25(OH)_2_D_3_ suppressed estrogen-induced proliferation in leiomyoma cells [50]. Cell proliferation and extracellular matrix production in myoma tumors can be affected by Vitamin D as it can suppress tumor-promoting Wnt4/β-catenin expression and reduce activation of mTOR signaling in human UF cells [82]. As observed by Corachán et al., Vitamin D inhibits the Wnt/β-catenin and TGFβ pathways, reducing proliferation and extracellular matrix formation, in different molecular subtypes of uterine myomas (MED12-mutated and wild-type human tumors) [83]. Ali et al. hypothesized that myoma tumor progression might be inhibited by recovering the damaged DNA repair system [84]. They showed in vitro that vitamin D_3_ treatment significantly reduces DNA damage, restores the normal DNA damage response, and is accompanied by induction of VDR in fibroid cells [84]. DNA repair in cells exposed to classic DNA damage inducers in UF pathogenesis (endocrine-disrupting chemicals -EDCs) was achieved by a 1,25(OH)_2_D_3_ treatment of myoma cells in animal models [85]. Clinical trials show promising results in patients with UFs and hypovitaminosis D treated with Vitamin D_3_ supplementation, revealing a significant decrease in tumor size and numbers [76,86]. At the time of the review, search results pointed to a randomized trial (RCT) being conducted in women at reproductive age affected with uterine myoma, aiming to evaluate whether supplementation with Vitamin D_3_ could reduce the risk and inhibit the growth of fibroids [87]. Results of the evaluation of Vitamin D_3_ effects in this particular group of women can be of value in everyday gynecological practice.

Paricalcitol, an analog of 1,25(OH)_2_D_3_, has less calcemic activity and, therefore, appears to be safer in long-term use than 1,25(OH)_2_D_3_. Halder et al. study indicates that treatment with paricalcitol has an inhibitory effect on uterine fibroid cell proliferation [69]. On a murine model, both paricalcitol and 1,25(OH)_2_D_3_ significantly reduced fibroid size, but paricalcitol was more potent. Porcaro et al. observed a significant reduction in myoma volume and overall improvement in quality of life in patients treated with a combination of Vitamin D, EGCG, and vitamin B6 [88]. This combined supplementation treatment presents as quite an innovative approach to treating leiomyoma with oral supplementation. Shen et al., on the other hand, arrived at contrary conclusions. In their study, Vitamin D supplementation had no effects on the risk of UFs [15].

A study by Güleç et al. determined the association between Vitamin D receptor polymorphisms and the occurrence of uterine myomas [89]. It shows that among the fok1 polymorphisms of the vitamin D receptor, the presence of the CC fok1 genotype may be a risk-reducing factor, and the T allele may increase the risk of uterine myomas. A recent study by Fazeli et al. evaluated CYP24A1 gene expression in uterine myoma tissue [90]. CYP24A1 is a mitochondrial enzyme that catalyzes the degradation of 1,25(OH)_2_D_3_ to its less active 25-D3 form and regulates the amount of active Vitamin D in tissues. The expression of CYP24A1 in leiomyoma suggests that local degradation of 1,25(OH)_2_D_3_ may also have a role in fibroma development.

Overall, Vitamin D_3_ may be a promising option in prevention and treatment of UFs. The majority of presented studies consider treatment with Vitamin D_3_ as safe and effective. 

### 2.5. Vitamin C

Vitamin C (ascorbic acid) is an antioxidative nutrient that prevents against the effects of oxidative stress and has anti-inflammatory properties [22,46]. It cannot however be biosynthesized by the human organism [91]. The best source of this vitamin is a diet rich in fruits (currants, acerola, cherries, and citrus fruits) and vegetables (tomatoes, peppers, cabbage, broccoli, and spinach). Vitamin C supplements are also available in a variety of forms. Pleiotropic effects of vitamin C (antioxidant, anti-inflammatory, immune system support, cofactor in hormone biosynthesis, and microcirculation protector) may suggest its use in treatment of myomas. Sadly, there is not much data regarding this matter. 

Martin et al. noted that the risk of UFs increased with higher Vitamin C levels, but the association was not significant [18]. A study by Heinonen et al. demonstrated dysregulation of Vitamin C metabolism in leiomyoma of the *MED12* subtype—a common type of mutation in UFs [4]. Ascorbic acid was also used as bleeding prevention in surgical myoma treatments, but the conclusions were contradictory [92,93]. Pourmatroud et al. observed that Vitamin C administration can reduce blood loss during abdominal myomectomy; Lee et al. did not confirm that in women undergoing laparoscopic myomectomy.

To conclude, there is very little data on Vitamin C supplementation in terms of its effects on leiomyoma, but higher consumption of fruits and vegetables may reduce the risk of myoma incidence.

### 2.6. Other Vitamins

Apart from two studies, there is virtually no data on the effects of other vitamins on the formation and management of UFs. The literature showed no clear associations between Vitamin B6, Vitamin B12, folate, and the occurrence of UFs [16,18].

## 3. The Active Compounds from Plants

### 3.1. Green Tea—Polyphenols

Green tea is widely known for its antioxidant activity and is extensively consumed, especially in Asian countries. Compared to other beverages, it has a much higher catechin content. Components of green tea include polyphenols (epigallocatechin-3-gallate—EGCG, epigallocatechin—EGC, epicatechin-3-gallate—ECG, epicatechin—EC), flavones, and flavanols (kaempferol, myricetin, quercetin). The average daily intake of EGCG from green tea consumption in the EU ranges from 90 to 300 mg/day, while high-level consumers intake even up to 860 mg EGCG/day. In vitro studies showed that EGCG, consumed in the form of a green tea extract, inhibited proliferation and growth and promoted apoptosis in cultures of human uterine leiomyoma cells in a dose-dependent manner [94,95]. Antiproliferative and gene-modulating effects of EGCG were partially mediated through the effect on catechol-O-methyltransferase (COMT) enzyme activity. Similar EGCG action effects were observed in vivo in animal models [96,97]. Ozercan et al. observed that EGCG extract decreased tumor necrosis factor α (TNFα) levels, a cytokine associated with leiomyoma pathophysiology [97]. It appears that EGCG supplementation, by modulating multiple cellular signaling pathways, reduces tumor size and may be an alternative therapeutic option in treatment of UFs. Roshdy et al. evaluated green tea extract (EGCG) taken orally as safe and effective treatment for symptomatic UFs [98]. EGCG intake at a dose of 800 mg/day resulted not only in a significant reduction in tumor volume but also in a reduction of fibroid-specific symptoms, and many treated patients experienced improved health-related quality of life. However, the European Food Safety Authority (EFSA) notes the potential adverse hepatotoxic effects of green tea catechins at intakes ≥ 800 mg EGCG/day taken as a dietary supplement [99]. Grandi et al. observed that EGCG at a daily dose of 300 mg, when vitamin B6 and vitamin D were added, significantly reduced the volume of intramural (mainly) and subserosal UFs [100]. The 90-day treatment resulted in a significant reduction in the length of menstrual bleeding, but not significant changes in health-related quality of life or an improved comfort of sex-life. Grandi et al. suggest EGCG supplementation as an alternative method of treatment for women in late reproductive age, when hormone therapy is not optional. Similar results were observed by Porcaro et al. for a dose of 150 mg EGCG with 25 μg vitamin D and 5 mg vitamin B6 intake in women at reproductive age with symptomatic myomas [88]. Young women at childbearing age can also benefit from EGCG supplementation. Miriello et al. noted that combined daily supplementation of EGCG (300 mg), vitamin D (50 μg), and vitamin B6 (10 mg) can, with no side effects, reduce myoma volume and related symptoms and improve patients’ quality of life [101].

EGCG under normal, physiological conditions is characterized by low stability, poor bioavailability, and high metabolic changes. Therefore, methods are being sought to improve the stability of EGCG as a drug. Ahmed et al. studied the biological properties of pro-drug EGCG analogs (pro-EGCG analogs) in human leiomyoma cell lines [102]. They found that these drugs, with improved stability, bioavailability, and biological activity, exhibited potent antiproliferative, antiangiogenic, proapoptotic, and antifibrotic activities in UFs. Pro-drugs EGCG analogs share the same molecular targets as natural EGCG in inhibiting enzymatic activity and could potentially be more effective than natural EGCG therapeutic agent in a long-term use in women with symptomatic UFs. Contrary results were reported in an observational study by Biro et al. [103]. They found that consumption of green tea extract (GTE) capsules resulted only in a significant improvement in physical quality of life (QoL) score. No changes were observed in myoma size or myoma-related complaints or in global QoL score after GTE supplementation. Shen et al. arrived at the same conclusion: that drinking green tea had no effect on the risk of leiomyoma [15].

To sum up, the effects of UFs treatment with polyphenols can depend on dose, duration of treatment, and patient selection.

### 3.2. Curcumin/Turmeric

Curcumin is one of the three major curcuminoids in turmeric plant (*Curcuma longa*). Numerous studies have highlighted its antioxidant, anti-inflammatory, anti-carcinogenic and immunoregulatory activity at the molecular level [104,105,106]. By suppressing anti-apoptotic proteins, curcumin can protect against the formation and growth of tumors, including uterine myomas.

Malik et al. demonstrated in vitro that curcumin inhibited the proliferation of uterine leiomyoma cells [107]. The curcumin compound caused upregulation of the apoptotic pathway and inhibited fibronectin production, a component of ECM [107]. Tsuiji et al. noted the effect of curcumin on peroxisome proliferator-activated receptor-gamma (PPARγ) activation [108]. Feng et al. demonstrated in an animal model that Rhizoma Curcumae (RC) and Rhizoma Sparganii (RS), used in traditional Chinese medicine, were effective in preventing and treating UFs in rats [109]. The combination of RC and RS effectively reduced the expression of extracellular matrix component collagen, fibroblast activating protein, and transforming growth factor beta (TGF-β), simultaneously decreasing the expression level of signaling factors (AKT, ERK and MEK) in cell proliferation [109]. Similar effects were observed by Yu et al. [110]. They noted that RC/RS herbs regulated key pathways in UF cell proliferation and ECM formation, such as MAPK, PPAR, Notch, and TGF-β/Smad. Curcumin can thus reduce the oxidative stress and protect against inflammation. This activity is expressed through modulation of proinflammatory cytokines and signaling pathways, including beforementioned peroxisome proliferator-activated receptor gamma (PPAR-γ) [105].

In conclusion, turmeric appears to be a desirable dietary component for women at risk of developing uterine myomas and those already affected by this disease.

### 3.3. Quercetin and Indole-3-Carbinol

Indole-3-carbinol (I3C) and quercetin, as plant- derived components, were also of interest in a potential treatment of UFs. The former one can be sourced mainly from cruciferous vegetables, the latter is an active compound of onion. A recent in-vitro study revealed anti-fibrotic and anti-migratory effects of quercetin and I3C for uterine leiomyomas, but with no impact on myoma cell proliferation [111].

## 4. Micro- and Macro Elements

### 4.1. Selenium

Selenium (Se) plays an integral part of ROS detoxifying selenoenzymes such as glutathione peroxidases and thioredoxin reductases, and that is why its antioxidant function can potentially have an effect on uterine leiomyomas [112]. The expression of selenium-binding protein 1 was found to be decreased in uterine leiomyomas. The role this protein plays in tumorigenesis was highlighted, and selenium intake was indicated for the prevention and treatment of uterine leiomyomas. Tuzcu et al. showed in animal models that dietary selenium supplementation reduces the size of spontaneously occurring leiomyoma [113]. The selenium-rich diet had no effect on the number of tumors.

### 4.2. Other Trace Elements

There appears to be only very few studies concerning the effects of trace elements on uterine myomas. In Nasiadek et al.’s study, cadmium (Cd) concentration was significantly lower in myoma than in the surrounding muscle. A significant increase in magnesium (Mg) and magnesium/calcium (Mg/Ca) ratio, with simultaneous decrease in iron (Fe) were also found in tumor tissue [114]. This may be associated with dairy consumption. Makwe et al. found that in African women at reproductive age suffering from leiomyoma, serum levels of magnesium (Mg) and phosphorus (P) were not significantly different when compared with healthy women at reproductive age [22]. Johnstone et al. showed that higher blood concentrations of cadmium (Cd) and lead (Pb) combined with higher urinary levels of cobalt (Co) could be positively associated with the higher risk of myoma incidence. They argued that increased exposure to these trace elements may stimulate fibroid growth and that fibroid tumors may act as a tissue reservoir for these elements [115].

In summary, of the dietary trace elements, only selenium has proven efficacy in preventing and treating uterine myomas. Heavy metals that can be transferred to food (i.e., from food containers) increase the risk of myoma and stimulate fibroid growth.

## 5. Endocrine-Disrupting Chemicals

It may be worth noting that some environmental contaminants can interfere with the beneficial effects of certain foodstuffs. These contaminants are known as endocrine-disrupting chemicals (EDC). By binding to hormone receptors, EDCs can stimulate these receptors and alter the function or production of natural hormones. Induction of both genomic and non-genomic signaling and pro-inflammatory effects of EDCs increase the risk of UFs [116]. Butylated hydroxytoluene (BHT), one of the most widely used antioxidant additives, can improve the stability of fat-soluble vitamins and prevent food spoilage. Results of recent in vitro studies suggest detrimental effects of BHT exposure on uterine myoma progression: increased cell proliferation, colony formation, and ECM accumulation [117]. Polychlorinated biphenyls (PCBs), a class of environmental pollutants found in fruits and vegetables, are also known to be endocrine disruptors. The potential risks of PCBs are associated with the consequences of PCBs binding to estrogen receptors. In their in vitro study, Wang et al. showed that Bisphenol A (BPA) in particular could increase cell proliferation and colony formation, and thus promote tumor growth [118]. Other studies, however, found no correlation between exposure to PCBs and UFs [119,120].

Despite the beneficial effects of food on myoma, the additives contained in food may nullify its positive impact and promote the formation and growth of fibroids.

## 6. Conclusions

Fibroids are fibrotic tumors with extracellular matrix deposition. Oxidative stress affects the formation and growth of these tumors [8]. The quality of diet can impact the formation and progression of fibroids and, as such, offer an innovative approach to treating these tumors. The micronutrient-uterine fibroids correlation may potentially be modified by ethnicity. Positive and negative nutritional effects on myoma are presented below in Figure 1.

The link between Vitamin D deficiency and the risk of UFs was strongly indicated. Vitamin D deficiency can stimulate cell proliferation and leiomyoma growth [61]. Physiological vitamin D concentrations effectively inhibit fibroid cell growth, especially in patients with hypovitaminosis D. The active form of Vitamin D has an affinity for binding to Vitamin D receptors (VDR) expressed in uterine myoma tissue. In fact, Vitamin D_3_ appears to be an ideal, inexpensive therapeutic agent for non-invasive treatment of UFs, or at least fibroid growth stabilization, without contraindications and side effects of hormone therapy. So far, the combination of ulipristal acetate and vitamin D_3_ treatment for UFs has not been extensively explored in the literature [121,122]. The use of Vitamin D_3_ analogues can eliminate the degradation of Vitamin D_3_ by tissue 24-hydroxylase, while maintaining the beneficial effects of Vitamin D_3_ on UFs [69]. Vitamin D_3_ can also promote a healthier composition of the intestinal microbiota, which improves the gut barrier function [123].

The literature findings about the role Vitamin A plays in uterine myoma are contradictory. It has been, however, indicated that treatment with retinoids reduces cell proliferation and ECM formation and increases apoptosis in fibroids [37,40,41]. A diet rich in Vitamin A, as well as the use of synthetic retinoid analogs, can prevent the development of fibroids and limit their growth. A diet rich in carotenoids, such as lycopene, can reduce the incidence of UFs and promote decrease in volume [14,15,16,17,35,36].

Based on the available studies, no clear conclusions can be drawn regarding vitamin E and C supplementation. Researchers emphasize the role of alpha-tocopherol, as phytoestrogen, in the modulation of estrogen receptors (ERs) [47]. ERs and estrogens are proven to be involved in the etiology of fibroids.

A daily diet enriched in fruits and vegetables, which are known sources of carotenoids, quercetin, and indole-3-carbinol, may be one of the simplest and modifiable lifestyle changes with beneficial effects in patients affected by leiomyoma. EGCG and green tea extract may also have applications in the prevention and treatment of UFs [15,94,96,98,123]. A combination of natural substances and vitamins i.e., Vitamin D and EGCG, has been evaluated as useful and effective in reducing fibroid-related symptoms and improving quality of life [88,100]. It can be applicable in patients with contraindications to hormonal treatment or those who wish to avoid surgical treatment altogether. Curcumin, another natural substance with proven therapeutic effects on UFs, can reduce oxidative stress and protect against inflammation in leiomyoma. Its action is expressed through modulation of pro-inflammatory cytokines and signaling pathways, including peroxisome proliferator-activated receptor-gamma (PPAR-γ) [105,107,108,109,110].

Very few studies considered the effects of probiotics on UFs. However, a protective role of dairy products, including yogurt consumption, has been pointed out [13,15,17,20,21].

To date, there have been only few studies evaluating the potential effects of trace elements on UFs [22,112,113,115]. Selenium, in particular, has shown to have protective properties against oxidative damage and inflammation and has been indicated for supplantation in leiomyoma [112,113].

Using natural compounds in treatment of UFs appears to be a worthwhile endeavor. Natural compounds present as an alternative route in UF treatment, especially in patients with contraindications for hormonal therapy. In women treated conventionally, natural compounds can strengthen therapeutic effects.

## 7. Methods

### Search Strategy

This study is based on an analysis of available studies focusing on correlations between uterine fibroid risk and treatment and diet. The aim of the review was to evaluate data on natural, non-hormonal, effective, and safe therapeutic options for treatment of this disease. An extensive search of PubMed and Medline resources was conducted, aiming at literature published between 2000 and 2021. A combination of the following phrases was used: “uterine fibroids”; “uterine myoma”; “leiomyomata”; “diet”, “fruits”; “vegetables”; “plants”; “dairy products”; “curcumin”; “turmeric”; “green tea”; “selenium”; “carotenoids”; “vitamin D”; “Vitamin C”; “Vitamin E”; “Vitamin A”; “dysbiosis”; and “gut microbiota”. All retrieved articles (*n* = 272), collected through the e-search process, were then reviewed by two researchers. Publications were limited to the English language. Literature unrelated to the review theme or replicated in database and conference abstracts were excluded from the analysis. Only full-text studies fulfilling the relevant, above-outlined criteria were included in the review. Approximately 120 publications were evaluated.

## Figures and Tables

**Figure 1 nutrients-14-00734-f001:**
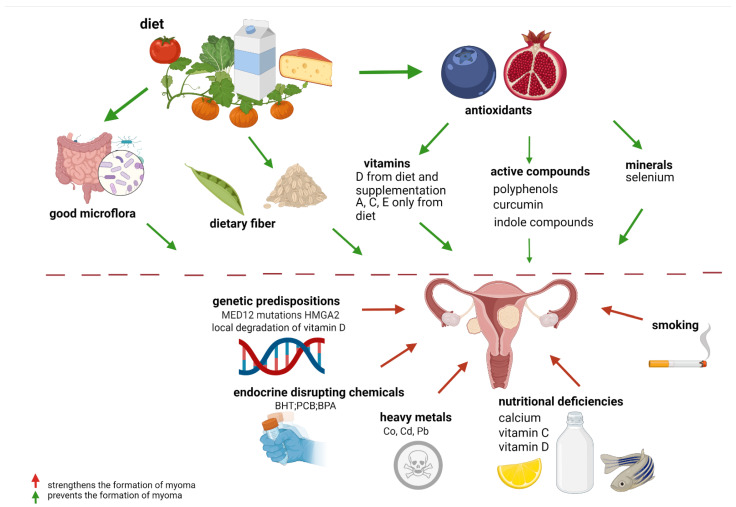
Effects of nutrients and environmental factors on leiomyoma. Legend: MED 12 mutations—mediator complex subunit 12; HMGA2—high mobility group AT-hook 2.; BHT—butylated hydroxytoluene; PCB—polychlorinated biphenyls; BPA—bisphenol A; Co—cobalt; Cd—cadmium; Pb—lead. (Created with BioRender.com https://app.biorender.com/, accessed on 30 November 2021).

## Data Availability

The study did not report any data.

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
