# Peer review of "Dietary Natural Compounds and Vitamins as Potential Cofactors in Uterine Fibroids Growth and Development"

_nutrients, 2022, doi:10.3390/nu14040734_

Round 1

Reviewer 1 Report

General comment: The authors presented an interesting review work concerning the role of diet in uterine fibroids growth and development.

The manuscript is written in a comprehensive way.

Title: The title is short, concise, and adequate.

Abstract: It is adequate.

The keywords should be different from those used in the title.

The manuscript is well structured.

Introduction: It is adequate. The authors provided an adequate overview of the thematic.

Once defined, the abbreviations should be used throughout the manuscript. Please delete full name “uterine fibroids” (page 2, line 85).

The studies provided in Table 1 should be presented by chronological order.

Methods: The methods are well described.

Conclusion: The conclusion is based in the results of the studies included in the review.  

Reviewer 2 Report

The study falls whitin journal’s aim.

Despite good overall merit, I suggest to add tables showing

details of the studies included in the narrative review (population, outcomes, etc

Please highilight also evidence level of each stusy
